# Mimicking Native Display of CD0873 on Liposomes Augments Its Potency as an Oral Vaccine against *Clostridioides difficile*

**DOI:** 10.3390/vaccines9121453

**Published:** 2021-12-08

**Authors:** Cansu Karyal, Panayiota Palazi, Jaime Hughes, Rhys C. Griffiths, Ruby R. Persaud, Patrick J. Tighe, Nicholas J. Mitchell, Ruth Griffin

**Affiliations:** 1Synthetic Biology Research Centre, School of Life Sciences, The University of Nottingham Biodiscovery Institute, Nottingham NG7 2RD, UK; cansu.karyal@nottingham.ac.uk (C.K.); jaime.hughes@nottingham.ac.uk (J.H.); 2School of Chemistry, University of Nottingham, Nottingham NG7 2RD, UK; panayiota.palazi@nottingham.ac.uk (P.P.); rhys.griffiths@nottingham.ac.uk (R.C.G.); nicholas.mitchell@nottingham.ac.uk (N.J.M.); 3Institute of Immunology and Immunotherapy, College of Medical and Dental Sciences, The University of Birmingham, Birmingham B15 2TT, UK; MRP813@student.bham.ac.uk; 4School of Life Sciences, The University of Nottingham, Nottingham NG7 2RD, UK; paddy.tighe@nottingham.ac.uk; 5NIHR Nottingham Biomedical Research Centre, Nottingham University Hospitals NHS Trust, The University of Nottingham, Nottingham NG7 2UH, UK

**Keywords:** oral vaccine, *Clostridioides difficile*, recombinant protein, sIgA, IgG, lipidation, liposomes

## Abstract

Mucosal vaccination aims to prevent infection mainly by inducing secretory IgA (sIgA) antibody, which neutralises pathogens and enterotoxins by blocking their attachment to epithelial cells. We previously demonstrated that encapsulated protein antigen CD0873 given orally to hamsters induces neutralising antibodies locally as well as systemically, affording partial protection against *Clostridioides difficile* infection. The aim of this study was to determine whether displaying CD0873 on liposomes, mimicking native presentation, would drive a stronger antibody response. The recombinant form we previously tested resembles the naturally cleaved lipoprotein commencing with a cysteine but lacking lipid modification. A synthetic lipid (DHPPA-Mal) was designed for conjugation of this protein via its N-terminal cysteine to the maleimide headgroup. DHPPA-Mal was first formulated with liposomes to produce MalLipo; then, CD0873 was conjugated to headgroups protruding from the outer envelope to generate CD0873-MalLipo. The immunogenicity of CD0873-MalLipo was compared to CD0873 in hamsters. Intestinal sIgA and CD0873-specific serum IgG were induced in all vaccinated animals; however, neutralising activity was greatest for the CD0873-MalLipo group. Our data hold great promise for development of a novel oral vaccine platform driving intestinal and systemic immune responses.

## 1. Introduction

Vaccination is the most effective medical intervention to prevent the spread of infectious diseases [1]. However, with the rise in antimicrobial resistant infections, communicable diseases are predicted to exceed cancer by 2050 and claim 10 million lives a year [2] (WHO, 2019). Enteric infections alone cause more than a billion disease episodes annually and claim nearly 2 million lives each year [3].

Most licensed vaccines are administered parenterally as intramuscular or subcutaneous injections. First-generation vaccines are composed of the whole organism, either live attenuated or killed [4]. However, attenuated vaccines pose a safety risk, which limits their use in the elderly or immunocompromised. Aside from the harmful pathogenic material they contain, live vaccines can undergo spontaneous mutations and revert to their infectious form, risking infecting the host [5,6]. To overcome this problem, efforts are focused on identifying individual antigens capable of safely eliciting immunoprotection [7]. However, subunit vaccines are generally poorly immunogenic, necessitating co-administration with immunostimulatory adjuvants [8,9]. A small number of adjuvants have been approved for injected vaccines, including aluminium salts, monophosphoryl lipid A, squalene-based oil-in-water emulsions and virosomes [10]. Of these adjuvants, alum salts are the most widely used; however, their immunostimulatory activity can be weak, and formulations containing alum cannot be stabilised by freeze-drying [11].

Intensive efforts have focused on identifying new adjuvants, and one strategy has been to add back microbial factors that elicit strong immune responses to subunit formulations. These factors are known as pathogen-associated molecular patterns (PAMPs) [12,13]. PAMPs are recognized as danger signals by pattern recognition receptors (PRRs) on the surface of antigen presenting cells (APCs) [7]. PAMPs activate PRRs such as Toll-like receptors (TLRs), which induce APCs to release cytokines and chemokines and to upregulate major histocompatibility complex (MHC) class II and costimulatory molecules. Professional APCs in turn prime T helper (Th) cells, which activate B cells to undergo class switching and produce antigen-specific antibodies. Furthermore, the fusion of PAMPs to antigens has been shown to significantly enhance antibody response compared to simply mixing the two together [10,14,15]. The reason for this is that conjugation of adjuvant to antigen ensures co-delivery to the same APC [16], which promotes optimal MHC class II presentation of the antigen and strong stimulation of Th cell responses [10,17]. An example of a natural antigen–adjuvant conjugate is a bacterial lipoprotein such as Factor H binding protein, which constitutes the meningococcal Trumenba vaccine developed by Pfizer.

Bacterial lipoproteins are characterised by their leader peptide, which ends in a lipobox with a conserved terminal cysteine [18]. A diacyl glyceryl group is conjugated to the thiol side-chain of cysteine through a thioether linkage, then the signal peptide is cleaved immediately prior to this cysteine (Figure 1A). In all Gram-negative bacteria and in certain Gram-positive bacteria, a third fatty acid is attached to this N-terminal cysteine through an amide linkage, generating a triacylated lipoprotein [19,20]. The lipid moiety anchors the lipoprotein to the membrane, orientating its protein component outwards from its N-terminus [20,21,22]. The lipid domain of lipoproteins is a PAMP which is recognised by TLR2 [23,24,25,26]. Examples of lipid moieties of bacterial lipoproteins that have been studied for their adjuvant activity are dipalmitoyl-*S*-glycerol cysteine, Pam_2_Cys (a synthetic version of the lipid moiety from macrophage-activating lipopeptide-2 derived from *Mycoplasma fermentans*, Figure 1A) and tripalmitoyl-*S*-glycerol cysteine, Pam_3_Cys, (a synthetic analogue of Brauns’ lipoprotein found in Gram-negative cell walls) [27,28,29,30].

Adjuvant is not the only requisite for vaccines to be effective. The route of immunisation is an equally important factor to ensure targeting of the relevant body site. Ninety percent of all infections occur in mucosal surfaces lining the gastrointestinal, respiratory and urogenital tracts, where the first line of defense is secretory IgA (sIgA) [31]; however, parenteral vaccines are largely limited to targeting pathogens that have already breached the mucosal barrier, via serum IgG [32]. The limitations of the parenteral approach to protect against non-invasive gut pathogens may partly explain the inability of Cdiffense (Sanofi) intramuscular toxoid vaccine to protect against *Clostridioides difficile* infection, which resulted in termination of its Phase III development. Mucosal vaccination, on the other hand, can trigger humoural and cellular mediated immune protection not only at mucosal sites, but also systemically [33].

For oral vaccines, potency is of paramount importance to overcome tolerance developed by the gut, which is constantly exposed to antigens; however, identifying effective and safe oral adjuvants has been extremely challenging [3]. It is not surprising, then, that all oral vaccines to date rely on the traditional whole cell approach (live attenuated or inactivated), capitalising on the potency afforded by the presence of all the PAMPs of the organism [3]. Only a handful of oral vaccines have been licensed, which collectively target four infectious diseases: typhoid, cholera, rotavirus and polio. The historic issue around their safety remains, and clearly a breakthrough is needed to develop safe subunit oral vaccine platforms with suitable adjuvants that can bypass degradation in the stomach and reach the main mucosal inductive site of the distal small intestine, the gut-associated lymphoid tissue (GALT) [34]. The bioavailability of oral vaccines is vital, as antigens must be successfully taken up by M cells (specialised epithelial cells) and transcytosed to underlying APCs in the Peyer’s patches of the GALT to stimulate protective immune responses [34].

We recently reported that the colonisation factor CD0873 of *C. difficile* given orally in enteric capsules to hamsters induced local and serum-neutralising antibody responses which afforded partial protection against infection with a hypervirulent strain [35]. The recombinant protein we tested resembles the mature, cleaved polypeptide of this lipoprotein, lacking lipid modification at the N-terminal cysteine. The limitations of protein alone as an oral subunit vaccine, aside from it propensity for degradation, is insufficient potency due to lack of adjuvant, as well as its tendency to form aggregates, which can result in epitopes escaping immune recognition. Furthermore, soluble protein antigens are not as immunogenic as particulate antigens at mucosal sites, which have greater propensity for M cell uptake and for APC targeting [36]. Moreover, membrane-bound protein antigens are more effective at eliciting B cell responses than soluble antigens [37,38,39].

With the above knowledge, the aim of this study was to test the hypothesis that oral delivery of the whole protein antigen, CD0873, displayed on the outer membrane of liposomal nanoparticles could stimulate enhanced antibody responses. A synthetic lipid bearing features of Pam_2_Cys was designed for conjugation of the protein via its N-terminal cysteine (Figure 1B) and to act as a linker for its surface display on liposomes. In contrast to the acylated glycerol headgroup of Pam_2_Cys, whereby the lipids are linked to the glycerol via ester bonds (Figure 1A), our lipid N-(2,3-bis(hexadecyloxy)propyl)-3-(2,5-dioxo-2,5-dihydro-1H-pyrrol-1-yl)propanamide (DHPPA-Mal) carries an alkylated amino-propanediol unit such that the lipids are linked via ether bonds for enhanced stability in the gut (Figure 1B). A maleimide moiety is coupled to the amine of this subunit to enable facile conjugation to the N-terminal cysteine (via a thioether bond) of the recombinant antigen (Figure 1B).

Liposomes are phospholipid membrane vesicles with intrinsic adjuvant properties that have attracted considerable interest as mucosal delivery systems [40]. Several studies have shown that following oral administration, liposomes, like certain other nanoparticles, are readily taken up by M cells [41,42]. We anticipated that the outer envelope of a liposome, which resembles the phospholipid membrane of a bacterium, could be exploited to present antigens “natively”. Specifically, the synthetic lipid could be formulated with liposomal lipids and the whole protein antigen subsequently conjugated to headgroups protruding from the outer envelope. This would allow orientation of the lipoprotein outwards from the N-terminus, mimicking the native presentation of lipoproteins that are anchored to the bacterial membrane (Figure 1C,D). The composition of liposomes (Lipo) was based on the work of Han, who specifically optimised formulations for the gut [43]. Dipalmitoylphosphatidylcholine (DPPC) and dipalmitoylphosphatidylserine (DPPS) in combination are effective at targeting liposomes to macrophages, DPPS is additionally effective at inducing IgA, and cholesterol is included for stability [43,44,45]. DHPPA-Mal was formulated with Lipo to create MalLipo.

To test whether whole proteins could be stably conjugated to MalLipo, recombinant green fluorescent protein (GFP) was used as a test protein and all liposomal conjugations were analysed by Fluorescence-Activated Cell Sorting (FACS). Once verified, recombinant protein antigen CD0873 was conjugated to MalLipo to produce CD0873-MalLipo.

An immunogenicity study was conducted in hamsters to compare orally administered CD0873-MalLipo with CD0873 for antibody titres in the small intestine and in sera as well as for their respective neutralising properties. The Caco-2 cell line derived from human colorectal adenocarcinoma cells was chosen for all adherence assays, as this is the established intestinal epithelial model for *C. difficile* colonisation studies [46]. Our data show that lipidated CD0873 displayed on liposomes induces a greater neutralising antibody response than CD0873 given alone and causes no detectable immunopathology in the gut. Our modular platform can be applied to other vaccines targeting the intestine for which finding suitable delivery systems and adjuvants have been major bottlenecks.

## 2. Materials and Methods

### 2.1. Bacterial Strains

*Escherichia coli* strains “NEB^®^ 5-alpha competent *E. coli*” and “T7 Express competent *E. coli*” were used for cloning purposes recombinant protein expression, respectively. Both strains were purchased from New England Biolabs (NEB), Hitchin, UK. The strains were cultured aerobically in Luria Bertani (LB) broth with shaking or on LB agar (Fisher Bioreagents, Loughborough, UK) at 37 °C, unless stated otherwise. Where appropriate, ampicillin was added to a final concentration of 100 µg/mL.

*Clostridioides difficile* strain 630 was kindly provided by Peter Mullany, UCL. Strain 630 is a virulent, multi-drug resistant strain isolated in 1985 from a hospital patient with severe pseudomembranous colitis which spread to other patients on the same ward in Zurich, Switzerland [47]. This outbreak strain harbours the two toxins TcdA and TcdB and belongs to the PCR ribotype 012, and has now been adopted as the reference strain for laboratory studies [48].

*C. difficile* strain 630 was cultured in Brain Heart Infusion medium (Oxoid) supplemented with 5 μg/mL yeast extract and 0.1% (*w*/*v*) L-cysteine (BHIS) containing selective supplements, 250 μg/mL D-cycloserine and 8 μg/mL cefoxitin (Oxoid) (BHIS CC). The strain was incubated overnight at 37 °C in an anaerobic workstation (Don Whitley Scientific, Bingley, UK) with an atmosphere of CO_2_ (10%), H_2_ (10%) and N_2_ (80%).

### 2.2. Chemical Reagents

Commercially available reagents and reagent-grade solvents were purchased from Merck, Fluorochem or Fisher and used as received. Anhydrous solvents were purchased from SigmaMerck Group, Feltam, UK, with the exception of Tetrahydrofuran (THF) and dichloromethane (DCM), which were freshly distilled. All aqueous solutions were prepared using deionised water. Dry solvents were used when indicated in the procedure. Glassware was dried at 100 °C in a vacuum oven for 24 h.

### 2.3. Molecular Manipulations

Plasmid DNA was prepared using the Monarch^®^ Plasmid Miniprep Kit according to the manufacturer’s instructions (NEB). Enzymes for DNA manipulations included restriction endonucleases (NEB), T4 DNA ligase (NEB) and Calf Intestinal Alkaline Phosphatase (Invitrogen™, Thermo Fisher Scientific, Loughborough, UK). Gene cleaning was performed using the Monarch^®^ PCR and DNA Cleanup Kit (NEB) following the manufacturer’s instructions. DNA polymerase used for PCR for cloning purposes was Q5^®^ High-Fidelity DNA Polymerase (NEB), or *Taq* DNA polymerase (NEB) for verification of constructs. PCRs were performed in an Eppendorf^®^ Mastercycler^®^ (Stevenage, UK). Reaction mixtures were typically subjected to initial denaturation at 94 °C for 5 min followed by 35 cycles, denaturation at 94 °C for 30 s, annealing at the appropriate temperature for the primers for 30 s and extension at 72 °C for 30 s to 1 min depending on the length of amplicon, followed by a final extension cycle at 72 °C for 5 min. The amplified DNA was visualised by electrophoresis with 1% (*w*/*v*) agarose gels.

### 2.4. Cloning GFP Gene String into pTWIN1-His

Recombinant GFP was produced in the same manner as that described for recombinant CD0873 [35]. Briefly, the Intein Mediated Purification with an Affinity Chitin-binding Tag-Two Intein (IMPACT-TWIN) system was used (NEB), with our own modified pTWIN1-His vector [35]. The nucleotide sequence of cysteine-free GFP [49] was codon-optimised for *E. coli*, chemically synthesised (Life Technologies, Thermo Fisher Scientific), and then used as template for PCR with primers GFP For: 5′-GTGGTTGCTCTTCCAACTGC-3′ and GFP Rev: 5′-GGTGGTCTGCAGCTTGTACA-3′. The PCR products were digested with *Sap*I and *Pst*I and ligated into the *Sap*I-*Pst*I sites of pTWIN1-His, and ligation mixtures were used to transform NEB^®^ 5-alpha cells. Clones were confirmed by sequencing and designated pTWIN1-His-GFP, then transformed into T7 Express cells.

### 2.5. Expression and Purification of GFP by Immobilised Metal Affinity Chromatography (IMAC)

First, 1 L *E. coli* broth cultures A_600_ 0.6–0.7 were induced by adding isopropyl β-d-1-thiogalactopyranoside (IPTG) to a final concentration of 0.3 mM with shaking at room temperature overnight. Cells were harvested by centrifugation. The cell pellet was resuspended in ice-cold binding buffer (20 mM Tris-HCl, 1 M NaCl, 40 mM imidazole, pH 7.4), sonicated using the Fisherbrand™ Q500 Sonicator, then centrifuged. The supernatant (cytosolic/soluble fraction) was harvested, the pellet (insoluble fraction) re-suspended in binding buffer, and the fractions analysed by sodium dodecyl sulphate polyacrylamide gel electrophoresis (SDS-PAGE).

The soluble fraction was passed through a pre-charged Ni^2+^ PD-10 column (GE, Healthcare Life Sciences, Amersham, UK) containing HisPur Ni-NTA Resin (Thermo Scientific). The flow-through was collected, then the beads were washed with binding buffer to remove unbound proteins. The target protein was eluted with increasing concentrations of imidazole (50 mM, 100 mM, 250 mM and 500 mM) in elution buffer (0.1 M Tris-HCl and 2.5 M NaCl). All eluates were checked by SDS-PAGE, and those containing pure protein were combined for dialysis in PBS at 4 °C overnight.

### 2.6. Synthesis of Mal Lipid, N-(2,3-bis(hexadecyloxy)propyl)-3-(2,5-dioxo-2,5-dihydro-1H-pyrrol-1-yl)propanamide) Designated DHPPA-Mal (***4***)

#### 2.6.1. (*S*)-2,3-bis(hexadecyloxy)propan-1-amine (**3**)

A solution of cetyl alcohol (5.0 g, 20.6 mmol) and Et_3_N (4.16 g, 41.1 mmol) in anhydrous DCM (100 mL) was cooled to 0 °C before mesyl chloride (2.84 g, 24.8 mmol) was added dropwise. The solution was allowed to warm to room temperature and stirred for 16 h. After this time the reaction solution was diluted with DCM (100 mL) and washed with 2 M HCl, 1 M NaHCO_3_, and brine. The organic extract was dried over MgSO_4_, filtered and concentrated *in vacuo*. The resulting yellow solid was dried under high vacuum for 4 h, yielding pale yellow flakes (**1**, 6.34 g, 19.8 mmol, 96% yield). HRMS Calc.: 343.2283 [M + Na]^+^; Obs.: 343.2276 [M + Na]^+^. This product, (**1**), was taken on to the next step without further purification.

A solution of (*S*)-3-amino-1,2-propanediol (0.23 g, 2.5 mmol) in anhydrous DCM:MeOH (10:1, 25 mL) was stirred over Na_2_SO_4_ (1.78 g, 12.5 mmol) for 2 h before the drop-wise addition of benzaldehyde (0.27 g, 2.5 mmol). The mixture was stirred at room temperature for 24 h before filtration of the solid and concentration of the solution *in vacuo*. The resulting oil (**2**) was dried under high vacuum for 6 h before being used directly. To an oven dried and argon (Ar) flushed RB (round bottom) flask was added NaH (0.50 g, 12.5 mmol, 60% dispersion in mineral oil, 5 equiv.). To this was added a solution of **2** (0.45 g, 2.5 mmol, 1 equiv.) in anhydrous THF (10 mL) under an Ar atmosphere. This mixture was stirred at room temperature for 1 h before a solution of crude **1** (4.02 g, 12.6 mmol, 5 equiv.) in anhydrous THF (10 mL) was added dropwise. Once added, the solution was heated to reflux and left for 72 h under Ar. The reaction was quenched by the addition of water (100 mL) and the aqueous layer extracted with DCM. The combined organic extracts were washed with water, washed with brine, dried over MgSO_4_, filtered, and concentrated *in vacuo*. The resulting oil was dissolved in DCM:TFA (1:1, 100 mL) and stirred at room temperature for 16 h. Reaction progress was monitored by TLC. The reaction was concentrated *in vacuo* and the TFA co-evaporated with toluene. The resulting material was dried under high vacuum to yield a brown oil, which was purified by column chromatography (DCM to DCM:MeOH 8:2), yielding a brown solid (**3**, 0.57 g, 1.05 mmol, 42% yield). HRMS Calc.: 540.5720 [M + H]^+^; Obs.: 540.5751 [M + H]^+^. ^1^H NMR (400 MHz, methanol-d4) δ 3.61–3.55 (m, 2H), 3.53–3.47 (m, 3H), 3.43 (t, J = 6.7 Hz, 2H), 3.08 (dd, J = 13.0, 4.3 Hz, 1H), 2.98 (dd, J = 13.1, 6.6 Hz, 1H), 1.59–1.55 (m, 4H), 1.31–1.25 (m, 52H), 0.88 (t, J = 6.8 Hz, 6H). ^13^C NMR (100 MHz, methanol-d4) δ 77.9, 77.6, 77.2, 70.3, 70.1, 69.8, 31.8, 29.7, 29.5, 29.5, 29.3, 29.2, 25.9, 25.9, 22.5, 13.6.

#### 2.6.2. (*S*)-*N*-(2,3-bis(hexadecyloxy)propyl)-3-(2,5-dioxo-2,5-dihydro-1H-pyrrol-1-yl)propanamide (DHPPA-Mal) (**4**)

To a solution of 3-maleimidopropionic acid (45 mg, 0.27 mmol) in DMF (5 mL) was added HATU (106 mg, 0.28 mmol) and DIPEA (71 mg, 0.55 mmol). This mixture was stirred for 5 min before the addition of **3** (150 mg, 0.28 mmol). The solution was left to stir for 24 h before being acidified with 1 M HCl, and the aqueous layer was extracted with EtOAc. The combined organic layers were washed with water, washed with brine, dried over MgSO_4_, filtered and concentrated *in vacuo*. The resulting oil was purified by column chromatography (DCM to DCM:MeOH 9:1) and dried under high vacuum to yield a brown solid (**4**, 50 mg, 0.072 mmol, 27% yield). HRMS Calc.: 690.6036 [M + H]^+^; Obs.: 691.5982 [M + H]^+^. ^1^H NMR (400 MHz, CDCl_3_) δ 6.69 (s, 2H), 5.96–5.94 (m, 1H), 3.84 (t, J = 7.3 Hz, 2H), 3.59–3.38 (m, 8H), 3.30–3.24 (m, 1H), 2.50 (t, J = 7.3 Hz, 2H), 1.55 (m, 4H), 1.3–1.25 (m, 52H), 0.88 (t, J = 6.8 Hz, 6H). ^13^C NMR (100 MHz, CDCl_3_) δ 170.4, 169.5, 134.20, 77.3, 77.0, 76.7, 76.4, 71.9, 71.5, 70.2, 40.9, 34.6, 34.3, 31.9, 30.0, 29.7, 29.7, 29.5, 29.4, 26.1, 22.7, 14.1.

### 2.7. Lipid Analysis

High-resolution mass spectra were recorded on a Bruker MicroTOF Focus II MS (ESI) operating in positive or negative ionisation mode. NMR samples were analysed on a Bruker AVIII 400 NMR system (^1^H-NMR frequency 400 MHz; ^13^C-NMR frequency 100 MHz). Chemical shifts are reported in parts per million (ppm) and are referenced to solvent residual signals: CDCl_3_ (δ 7.26 [^1^H]). ^1^H NMR data is reported as chemical shift (δ), multiplicity (s = singlet, d = doublet, t = triplet, q = quartet, dd = doublet of doublets, ddd = doublet of doublet of doublets or combinations of these multiples; m = unassigned multiplet), relative integral and coupling constant (J Hz).

### 2.8. Preparation of Liposomes

Liposomes were formulated with 3 µmol dipalmitoylphosphatidylserine (DPPS) (Avanti^®^ Polar Lipids, Alabaster, AL, USA), 3 µmol dipalmitoylphosphatidylcholine (DPPC) (Avanti^®^ Polar Lipids, Alabaster, AL, USA) and 4 µmol cholesterol (Sigma Merck Group, Feltam, UK) (total lipid concentration, 1 mM), and are referred to as “Lipo”. A similar suspension was made containing 2.5 µmol DPPS, 2.5 µmol DPPC and 4 µmol cholesterol formulated with 1 µmol of DHPPA-Mal at a final concentration of 1 mM, and is referred to as “MalLipo”. DPPC, DHPPA-Mal, and cholesterol were dissolved in chloroform and DPPS in chloroform:methanol (1:1). Each lipid was dissolved to a concentration of 1 mM and the lipids combined in the appropriate ratio. Each liposomal suspension was mixed thoroughly in a 25 mL round bottom flask. The solvent was then evaporated using a Rotavapour^®^ R-114 (Büchi, Suffolk, UK) with the flask half-immersed in a Waterbath B480 (Büchi, Flawil, Switzerland) at 40 °C. The resulting film layer of lipids was further dried under high vacuum for 4 h. The film layer was resuspended in 10 mL PBS and sonicated on ice, using a 6 mm tip probe at 30% amplitude with 10 s on pulse and 30 s off pulse for a total of 30 min, with the Fisherbrand™ Q500 Sonicator (500 W, 20kHz) (ThermoFisher Scientific, Walthem, MA, USA). The resulting suspension of liposomes was analysed for Z-average particle size and polydispersity index (PDI) by dynamic light scattering (DLS). DLS was performed on a Malvern Zetasizer Nano-ZS; 1 mL of sample (formulation in PBS) was analysed in a disposable cuvette at 25 °C. Standard parameters were applied: material refractive index = 1.45; viscosity of PBS solution = 0.8872 cP; absorbance 0.100. Five measurements were taken per sample, each involving fourteen scans at a 173° scattering angle.

### 2.9. Conjugation of GFP to MalLipo

Since the liposomes were of the size range for multilamellar particles, it was estimated that about 10–20% of the DHPPA-Mal formulated would protrude outwards, with the remainder pointing inwards. GFP was used as a test protein for conjugation. The aim was to saturate protruding DHPPA-Mal with an excess of GFP, which was tested by comparing two molar equivalents: 2:1 and 5:1 of GFP to DHPPA-Mal in MalLipo. The MalLipo and protein suspensions were concentrated to approximately 1.5 mL using Vivaspin 20 columns with 50 kDa and 10 kDa cut-offs, respectively (GE Healthcare Life Sciences). Conjugations were conducted in the presence of Tris-(2-carboxyethyl)phosphine (TCEP) (pH 7.5) at a 1:2 molar ratio of protein to TCEP. Incubations were performed with gentle shaking at 50 rpm overnight. Conjugates were purified by size exclusion chromatography (SEC) to remove unbound GFP and excess TCEP. Briefly, SEC was performed using an ÄKTA Pure Chromatography System with a Superdex 200 10/300 GL column (10 × 300 mm). Column equilibration was performed for at least two column volumes in equilibration solvent (filtered distilled water) and elution solvent (PBS) at a flow rate of 0.75 mL/min. The wavelength detector was set to 280 nm and fractions were collected manually. Unicorn 7.0 software was used to record chromatograms.

The concentrations of GFP in all liposomal preparations were compared to the starting concentration as assessed by a BCA assay. As a negative control for the conjugation, GFP was incubated with Lipo. All formulations were compared by FACS analysis (2.10).

To test if the integrity of GFP-MalLipo liposomes would be affected by lyophilisation (a prerequisite for encapsulation in gelatin capsules), volumes of GFP-MalLipo containing a total of 1 mg protein were mixed with 10× lipid mass of trehalose, snap-frozen in liquid nitrogen, then freeze dried in a FreeZone 4.50 L, −84 °C Benchtop Freeze Dryer operating at −78 °C with vacuum 0.125 mbar. The resulting powdered formulations were reconstituted in PBS to the original volume and re-analysed by FACS.

### 2.10. FACS Analysis of GFP-MalLipo Formulations

FACS was conducted to evaluate conjugations of GFP to liposomes. The Astrios EQ Cell sorter (Beckman Coulter-Life Sciences, Wycombe, UK) was used, and all outputs were analysed using Kaluza Analysis 2.1 software (Beckman Coulter-Life Sciences, Wycombe, UK). Liposomes and GFP were individually gated by their forward scatter (FSC) and side scatter (SSC) using a 488 nm laser. For liposomes, both Lipo and MalLipo suspensions were used for positive gating and for establishing GFP negative fluorescence. The median florescence (MFI) was used to determine the relative median fluorescence (rMFI) by the equation:$$\mathrm{rMFI}=\frac{\mathrm{MFI}\;\mathrm{of}\;\mathrm{liposomes}\;\mathrm{in}\;\mathrm{gated}\;\mathrm{population}}{\mathrm{MFI}\;\mathrm{of}\;\mathrm{liposomes}\;\mathrm{in}\;\mathrm{negative}\;\mathrm{control}}$$

### 2.11. Conjugation of CD0873 to MalLipo, Lyophilisation and Capsule Packing

Conjugation of CD0873 to MalLipo was performed as described for GFP. The molar ratio of CD0873 to DHPPA-Mal was 1:3.8; 13.9 mM (total lipid concentration) was incubated with 20 mg of CD0873, with two equivalents of TCEP over protein. Following purification via SEC in PBS, 13 mg of CD0873 was successfully conjugated onto MalLipo, as verified by a BCA assay. Appropriate volumes of CD0873-MalLipo containing a total of 1 mg protein were aliquoted and lyophilised as described for GFP (2.9). Each powdered aliquot was packed separately into gelatin capsules, size 9 (Torpac Fairfield, NJ, USA) using the funnel, tamper and stand provided by the manufacturer. Capsules were dip-coated once in enteric polymer: 12.5% EUDRAGIT L100 (Evonik Industries, Essen, Germany) in isopropanol, with Triethyl citrate (TEC) (10% *w/v*) and H_2_O (3% *v/v*) added, as previously described [35].

### 2.12. In Vivo Immunogenicity Study

Female Golden Syrian hamsters aged 12–16 weeks, weighing approximately 150 g were purchased from Janvier Labs and housed in individually ventilated cages. Hamsters were randomly divided into four groups: Experimental groups were either given capsules containing CD0873 in excipient (*n* = 4) or CD0873-MalLipo in excipient (*n* = 4). Negative control groups were either given nothing (naïve) or capsules containing excipient only (*n* = 4). The purpose of this latter group was to check whether the capsule or excipient contributed towards any immunogenicity detected in experimental groups. Oral dosing with capsules was performed on days 1, 15 and 30. Hamsters were euthanised 14 days post-final immunisation. Blood was collected by cardiac puncture, left to clot overnight at 4 °C, and serum harvested after centrifugation and then stored at −80 °C. A 5 mm section of the ileum was taken for histological analysis (2.13). The remainder of the small intestine was placed in 5 mL PBS containing SIGMAFAST™ protease inhibitors (Sigma Merck Group, Feltam, UK), flushed through twice with this suspension, and the supernatant collected after centrifugation. Intestinal fluid was filter-sterilised and stored at −80 °C.

### 2.13. Histopathological Assessment of ileum

Sections of the ileum were fixed in 10% (*v*/*v*) neutral buffered formalin (NBF) (Sigma Merck Group, Feltam, UK) and then processed overnight on a 14 h programme in a TP1020 Automatic Benchtop Tissue Processor (Leica Biosystems, Newcastle, UK). Processed tissue was paraffin-embedded in a Histostar Embedding Workstation (Thermo Fisher Scientific) and sectioned at 5 µm, then mounted on glass slides and Hematoxylin and Eosin (H&E) stained (Leica ST 5020). Blinded analysis was conducted by an experienced pathologist using an established scoring system. Sections were assessed for oedema (0–3), neutrophil infiltration (0–3) and epithelial tissue damage (0–3), with 0 normal and 3 severe.

### 2.14. Western Immunoblotting

Procedures were performed with 5% (*w*/*v*) dry-milk (Sigma Merck Group, Feltam, UK) in Tris buffered saline containing 0.01% (*v*/*v*) Tween (TBST) for blocking. All antibodies were diluted in 1% (*w*/*v*) dry-milk in TBST, and TBST was used for washes. Samples were added to 2X Lammeli sample buffer and fractionated by 10% (*w*/*v*) SDS-PAGE. Transfer to PVDF membranes was conducted using the Trans-Blot Turbo Transfer System (Bio-Rad Laboratories, Watford, UK). For confirmation of recombinant proteins, the primary antibodies used were rabbit anti-His tag antibody (1:1000) (Cell Signaling Technology, CST, Danvers, MA, USA), mouse anti-GFP antibody (1:5000) (Roche diagnostics, Mannheim, Germany) and rabbit anti-CD0873 antibody (1:5000). Corresponding secondary antibodies were anti-rabbit IgG horseradish peroxidase (HRP) (1:1000) (CST) and anti-mouse IgG HRP (1:1000) (CST). Binding was detected by chemiluminescent ECL substrate and visualised using the LICOR Odyssey Fc (LICOR Biosciences, Cambridge, UK).

For detection of mucosal IgA in intestinal fluids, rabbit anti-hamster secondary antibody (Brookwood Biomedical, Jemison, Birmingham, AL, USA) was used which specifically detects IgA heavy chain (H). Briefly, anti-hamster IgA (H) antibody was purified from rabbit anti-hamster IgM, IgG, IgA (H) by cross absorption against hamster IgG and IgM (Brookwood Biomedical, Jemison, Birmingham, AL, USA) and used at 1:1000. Detection of bound antibody was achieved by incubation of the membrane with anti-rabbit IgG HRP antibody (1:1000) (CST) followed by 3,3′, 5,5′-Tetramethylbenzidine (TMB) substrate (Sigma Merck Group, Feltam, UK), and bands were visualised using the Gel Doc™ XR system (Bio-Rad, Hercules, CA, USA). Band intensity was quantified using ImageJ calibrated to perform optical density based on a pallet of colours in grayscale [50].

### 2.15. Indirect ELISA to Assess IgG Levels in Serum

Ninety-six-well Nunc MaxiSorp™ plates (Thermo Fisher Scientific, Loughborough, UK) were coated with 100 µL purified recombinant protein, CD0873, at a concentration of 2.5 µg/mL in 0.2 M sodium bicarbonate, pH 9.4 and the proteins left to adsorb onto the wells overnight at 4 °C. All wash stages consisted of five washes with 200 µL PBS containing 0.01% (*v*/*v*) Tween (PBST). Wells were first blocked with 200 µL of 5% (*w*/*v*) dry-milk (Sigma Merck Group, Feltam, UK) in PBST for 2 h at room temperature, washed and then incubated over night at 4 °C after addition of 100 µL serum diluted 1:10 in PBST, in triplicate. Wells were washed, then incubated for 2 h at room temperature in 100 µL goat anti-hamster IgG highly cross-adsorbed Biotin antibody (Sigma Merck Group, Feltam, UK) diluted 1:20,000 in PBST, washed again, then incubated for 2 h in Streptavidin-HRP (R&D Systems, Minneapolis, MN, USA) diluted 1:200 in PBST. *A*_650_ was measured after addition of 100 µL TMB substrate (Sigma Merck Group, Feltam, UK) for 15 min using the CLARIOstar Plus (BMG Labtech) Plate Reader.

### 2.16. Adherence Blocking Assay to Measure Neutralising Ability of Vaccine-Induced Antibodies

Caco-2 cells were cultured in Dulbecco’s Modified Eagles Medium (DMEM) (Thermo Fisher Scientific, Loughborough, UK) supplemented with 4.5 g/L D-glucose, 584 mg/L L-glutamine, 25 mM HEPES, 10% (*v*/*v*) FBS and penicillin/streptomycin in a humidified 5% CO_2_ atmosphere at 37 °C. Cells were seeded at 5 × 10^4^ cells per well in 24-well tissue culture plates (Corning, Flintshire, UK). Monolayers were used 14 days after seeding, with medium changed every 2–3 days. The culture medium was further changed 24 h prior to conducting the assay. The inoculum was prepared by standardising the optical density (OD) of a 10 mL overnight broth culture of *C. difficile* strain 630 to *A*_600_ 0.6, centrifuging and washing the cells with PBS, and re-suspending the pellet in non-supplemented DMEM. Caco-2 monolayers were infected at a multiplicity of infection (MOI) of 1:5 for intestinal fluid samples and 1:20 for serum samples, in triplicate. To confirm the MOIs, serial dilutions of the cell suspension in PBS were plated on BHIS CC for enumeration of CFUs. The adherence assay was performed under anaerobic conditions at 37 °C; 50 µL of serum or intestinal fluid diluted 1:5 and 1:2, respectively, in non-supplemented DMEM were added to 50 µL of the bacterial cell suspension and incubated for 1 h. This mixture was then added to Caco-2 cells in triplicate following removal of the medium in the wells. After 2 h of incubation, non-adherent bacteria were removed by pipetting and adherent bacteria harvested as follows. Caco-2 cells were washed three times with PBS, incubated in 200 µL 1X Trypsin-EDTA to detach them from the wells, then re-suspended in 300 µL supplemented DMEM. The following day, 10^−1^ to 10^−3^ dilutions of cells in PBS were plated on BHIS CC plates and CFU enumerated.

### 2.17. Statement

Animal studies were devised using the Experimental Design Assistant (EDA) online tool and conducted in strict accordance with the requirements of the Animals Scientific Procedure Act 1986. Prior approval for these procedures was granted by the University of Nottingham Animal Welfare and Ethical Review Body and by the UK Home Office under project license PPL P4712E8BB. Animals were euthanised by CO_2_ inhalation followed by cervical dislocation in order to minimize suffering.

### 2.18. Statistical Analysis

The data were analysed by Kruskal–Wallis test followed by Dunn’s multiple comparison. Error bars represent the standard error of the mean. * *p* < 0.05, ** *p* < 0.01, *** *p* < 0.001. All statistical tests were performed using GraphPad version 7 (San Diego, CA, USA); *p* values of less than 0.05 were considered to indicate statistical significance.

## 3. Results

The aim of this study was to test the hypothesis that oral delivery of the whole protein, CD0873, on liposomes resembling its display on bacterial membranes could induce a more effective antibody response than CD0873 alone. A synthetic lipid, DHPPA-Mal (Figure 1B), bearing features of the potent adjuvant Pam_2_Cys was designed for conjugation to the unique N-terminal cysteine of the recombinant protein. To mimic native display, DHPPA-Mal was first formulated with liposomal lipids. CD0873 was then conjugated to maleimide headgroups of this lipid protruding from the outer envelope of liposomes, mimicking the same orientation of lipoproteins as on bacterial cells (Figure 1C,D). To test the assembly of our nanoparticles, GFP bearing a unique N-terminal cysteine was used as a surrogate antigen. FACS was conducted to directly compare the fluorescence of GFP liposomes formulated with and without DHPPA-Mal.

### 3.1. Purification of Recombinant GFP with an N-Terminal Cysteine

The pTWIN1-His expression system was deployed to produce recombinant GFP with a unique N-terminal cysteine, as described previously for CD0873 [35]. When cloning into pTWIN1-His, the codon for cysteine is incorporated immediately after the *Sap*I cloning site in the forward primer, followed by the initial codons of the gene of interest. Following pH or temperature-mediated cleavage of the upstream intein tag, the cysteine forms the new N-terminus of the recombinant protein. A gene string was designed based on a previously modified GFP gene with two cysteine substitutions (C48S and C70M) that proved to be brighter and more photo-stable than standard GFP [49]. The nucleotide sequence of the cysteine-free gene was codon-optimised for *E. coli*, chemically synthesised and then cloned into the *Sap*I-*Pst*I sites of pTWIN1-His.

Following IPTG induction, the soluble fraction was applied to a Ni^2+^ column and GFP eluted with imidazole. A band of expected size for GFP (29 kDa) was obtained, as for previously purified CD0873 [35] and the two proteins confirmed by Western immunoblotting (Figure 2).

### 3.2. Synthesis of DHPPA-Mal

The bespoke lipid DHPPA-Mal (**4**) was synthesised as shown in Figure 1. Briefly, the amine of (*S*)-3-amino-1,2-propanediol was protected as an imine using benzaldehyde (**2**), before alkylation with mesylated cetyl alcohol (**1**) using NaH as the base. Hydrolysis of the imine under acidic conditions afforded amine **3** in 43% yield over two steps after purification via flash column chromatography. Then, 3-maleimidopropionic acid was coupled to the free amine of **3** using HATU as the coupling agent to afford DHPPA-Mal (**4**) in 34% yield after purification. NMR and MS data were consistent with the structure of compound **4** (DHPPA-Mal) (Figure A1, Figure A2 and Figure A3).

### 3.3. Formulation of Liposomes with DHPPA-Mal to Create MalLipo

Han and coworkers (1997) previously demonstrated that liposomes formulated with DPPC, DPPS and cholesterol (1:1:2 molar ratio) showed stability in acidic solution (pH2), bile and pancreatin solution, and were effective as an oral delivery vehicle for inducing mucosal immune responses to the encased antigen, in mice. With the need to include DHPPA-Mal to create MalLipo formulations, the ratio of lipids was adjusted accordingly and liposomes were formulated with 2.5 µmol DPPC, 2.5 µmol DPPS, 4 µmol cholesterol and 1 µmol DHPPA-Mal in a total volume of 10 mL PBS to create a 1 mM suspension (Figure 3). By thorough sonication, nanoparticles of 100–200 nm in diameter could be formed, which are of suitable size for uptake by M cells [41]. Analysis of the nanoparticles by DLS revealed that the Z-average particle size was 140 nm and the polydispersity index (PDI) was 0.20–0.26 (Figure A4). Lipo (liposomes lacking DHPPA-Mal) was formulated with 3 µmol DPPC, 3 µmol DPPS and 4 µmol cholesterol to yield a suspension of 1 mM in 10 mL.

### 3.4. Conjugation of Protein to Liposomes Using GFP a Test Protein with Analysis by FACS

GFP was used as a test protein for conjugation to MalLipo in order to assess conjugation efficiency; 2:1 and 5:1 molar ratios of GFP to DHPPA-Mal were used, and incubations were performed in TCEP (two equivalents over GFP) in order to reduce disulphide bonds and therefore minimise formation of aggregates due to protein–protein interactions. SEC was performed to remove unbound GFP. The efficiency of conjugation was initially assessed by BCA assay to compare the concentration of protein prior to conjugation and after SEC purification of conjugates. In order to first quantify the background level of association of GFP with liposomes as a result of electrostatic interactions, GFP was incubated with Lipo alone. Positive gating for liposomes was conducted on the Lipo and MalLipo suspensions (Figure 4A,B) which also served as negative gating for GFP fluorescence. GFP was positively gated on a GFP suspension (Figure 4C). All three suspensions showed an rMFI of 1 (Figure 4A–C). GFP incubated with Lipo, (GFP-Lipo) had an rMFI of 1.7 (Figure 4D), suggesting some GFP bound to liposomes electrostatically. GFP-MalLipo suspensions generated from a 2:1 and 5:1 molar ratio of GFP to DHPPA-Mal (Figure 4E,F) gave a 13-fold and 19-fold increase in the rMFI relative to GFP-Lipo with values of 21.5 and 32.5, respectively. The data show that GFP conjugated successfully to liposomes formulated with DHPPA-Mal.

In order to administer liposomal formulations orally, the selected commercial gelatin capsules require packing with dry powder formulations [35]. Thus, to test whether liposomal formulations could be lyophilised and reconstituted without affecting the structural integrity of the conjugated liposomes, aliquots of GFP-MalLipo suspensions, each containing a total of 1 mg of GFP, were lyophilised in 10X lipid mass of trehalose as excipient and then reconstituted in PBS to the original volume and reanalysed. No significant change in the fluorescence profile, particle size or PDI were observed, confirming the stability of the conjugates.

### 3.5. Conjugation of CD0873 to Liposomes

The same conjugation procedure were applied for recombinant CD0873; 20.0 mg of CD0873 was mixed with 13.9 mM MalLipo in TCEP in a total volume of 1.55 mL. Following purification via SEC, 13 mg of CD0873 were successfully conjugated onto MalLipo and verified by BCA assay. The nanoparticles were further analysed by DLS and displayed a Z-average size of 111 nm and a PDI of 0.32 (Figure A5). Appropriate volumes of CD0873-MalLipo containing a total of 1 mg protein were mixed with 10X lipid mass of trehalose and lyophilised. The powdered formulation was packed in Torpac capsules and the capsules dip-coated once in enteric polymer, as previously described [35], before oral administration to hamsters.

### 3.6. Induction of Mucosal IgA and Serum IgG in Hamsters Orally Immunised with CD0873 and CD0873-MalLipo

Hamsters were dosed orally, one capsule per dose, on days 1, 15, and 30, then euthanised two weeks later. One experimental group was immunised with capsules containing 1 mg of CD0873, and another group with capsules containing CD0873-MalLipo constituting 1 mg of CD0873. Negative control groups included naïve hamsters, and hamsters given capsules containing only excipient in order to verify that the capsule and/or trehalose had no effect. At the experimental endpoint, a section of the ileum was taken for histological analysis, and serum and intestinal fluid were harvested. Histological analysis revealed no oedema or increase in neutrophil infiltration or damage to the epithelium in any of the vaccinated hamsters, indicating good safety of the liposomal vaccine [35].

To investigate the prevalence of mucosal IgA in the small intestine, the diluted intestinal fluid of each hamster was probed by Western immunoblotting with rabbit anti-hamster IgA antibody and detected by anti-rabbit IgG HRP. An immuno-reactive band of the expected size [51] was observed for all hamsters vaccinated with either CD0873 or CD0873-MalLipo, unlike for the negative control groups. The intensity of each band was quantified using ImageJ and the results plotted for each group as a histogram. IgA abundance was greatest for the CD0873-MalLipo group, with *p* = 0.039 compared to the naïve group (Figure 5A). As with naïve animals, since only a background level of antibody was detected for the animals given capsules containing excipient only, this group was excluded from further analysis.

In order to test whether immunised hamsters generated a further systemic immune response, sera from immunised and naïve hamsters were compared by indirect ELISA. Wells were first coated in recombinant CD0873, then incubated in diluted serum. Biotin-labelled goat anti-hamster IgG was added, and Streptavidin-HRP was used for detection. Sera of both vaccinated groups contained significant levels of antigen-specific IgG compared to naïve animals: *p* = 0.0005 and *p* = 0.0065 for CD0873 and CD0873-MalLipo, respectively, with a 1.6-fold greater level for CD0873 compared to CD0873-MalLipo, *p* > 0.9999 (Figure 5B).

### 3.7. Induction of Greater Neutralising Antibody Responses by CD0873-MalLipo Compared to CD0873

To compare CD0873 and CD0873-MalLipo induced antibodies in terms of their neutralising properties, Caco-2 cells were infected with *C. difficile* strain 630 pre-incubated with intestinal fluid diluted 1:2 or sera diluted 1:5 from each animal. Two hours post-infection, cells were washed and detached from wells with Trypsin, serially diluted, plated, and the CFUs enumerated.

The intestinal fluid from both groups blocked adherence of cells of *C. difficile* to Caco-2 cells, with a significant reduction in adherence observed for the CD0873-MalLipo group compared to the naïve group (*p* = 0.0045), which was 1.9-fold greater than that observed for the CD0873 group (Figure 6A). The findings on the intestinal responses were mirrored by responses in sera. The sera of both groups significantly blocked adhesion, CD0873-MalLipo *p* = 0.0002 and CD0873 *p* = 0.0017, with the CD0873-MalLipo group exhibiting a 1.4-fold greater reduction in adherence than the CD0873 group (Figure 6B). Thus, despite the lower titre of IgG in the serum of the CD0873-MalLipo group (Figure 5B), the neutralising activity was again more marked for this group than for the CD0873 group.

To conclude, our data show that the presentation of whole protein antigen CD0873 on liposomes via conjugation with the DHPPA-Mal lipid to mimic native display of this lipoprotein on bacterial cells is more effective at inducing neutralising antibodies in the small intestine and bloodstream of hamsters than the recombinant antigen given alone.

## 4. Discussion

Mucosal vaccines have the potential to induce robust protective sIgA responses at the site of infection, preventing the initial establishment of disease as well as providing systemic IgG responses as a second line of defence. For protein-based vaccines, T-dependent B cell activation is required. The B cell receptor (BCR) must recognise its cognate antigen, and recognition is most effective if the antigen is encountered in its native form and is also membrane-bound [37,38,39]. In our study, we set out to test our oral protein antigen CD0873 [35] in this manner. Figure 7 illustrates the pathway from the initial transcytosis of antigen to the GALT, to recognition of the antigen by Follicular Th cells (CD4+ T cells) first on APCs and then on B cells, through to B cell activation, resulting in the secretion of antigen-specific sIgA.

Delivering whole protein resembling intact native antigen to the Peyer’s patches of the ileum is a major challenge. To overcome degradation in the stomach, we previously demonstrated the success of enteric capsules in targeting the small intestine of hamsters [35]. Our next strategy was to use robust, gut-optimised liposomes [43] to display membrane-bound antigen and deliver the antigen whole to the GALT. Traditionally, liposomes are deployed to display peptides as opposed to proteins [52]; however, this restricts the number of epitopes of the protein that can be presented and compromises their structural conformation, which can impact upon immune recognition. Our liposomes, which included cholesterol for solidity [43], provided the necessary structural support for whole proteins to remain stably conjugated, as demonstrated by the use of GFP. The composition of our liposomes was based on the formulation previously shown by Han, et al. (1997) [43] to be stable in gastro-intestinal fluids. The robustness of our liposomes likely afforded some protection of the antigen from degradation, enabling epitopes to be preserved. The individual conjugation of each CD0873 molecule to the synthetic lipid formulated with the liposomes would additionally serve to minimise protein aggregation, which can mask epitopes and affect recognition of naive B cells via their BCR.

Other benefits of our liposomal platform likely include improved uptake by M cells, as the lipid particles were of suitable size for uptake [41]. The combination of DPPC and DPPS in liposomes may have aided macrophage targeting following transcytosis. The enhanced biological activity of CD0873-MalLipo may be partly attributed to the liposomal lipid DPPS, which is known to have IgA-inducing properties [32], and to the synthetic lipid linker, which may have directly enhanced the immunogenicity of the attached antigen.

Pam_2_Cys consists of two palmitoyl fatty acid C16 chains adjoined to a glyceryl-cysteine motif by ester linkages (Figure 1A). Length-wise, C16 chains are optimal for activating cells through TLR2, while shorter chains induce no or little activity [53]. The ester linkages are also important for activating TLR2 [53]. However, synthesis of lipoproteins carrying this native scaffold is challenging, and therefore analogues that are easier to synthesise have been investigated instead for their immunostimulatory potential [53]. Important in the design of our vaccine platform targeting the intestine was the need for hydrolytic stability of the lipid and its facile conjugation to whole antigen. Retaining the core glyceryl (2,3-dihydroxypropyl) headgroup of Pam_2_Cys with native stereochemistry, which is substantially more active than the alternative (*S*)-diastereomer found in some analogues, we installed C16 chains via hydrolytically stable ether linkages. To enable effective bioconjugation of the antigen, we installed a maleimide moiety via a propanimidyl linker (Figure 1B). Facile conjugation of whole protein molecules to the outer envelope of MalLipo particles could then be performed under mild conditions between the N-terminal cysteine of the protein and protruding maleimide headgroups (Figure 1D and Figure 3B).

Our future work will be to deploy Pam_2_Cys itself, which is likely to be far more potent than DHPPA-Mal and is superior to other lipid PAMPs that have been tested. Moyle et al. (2014) compared Pam_2_Cys along with other lipids conjugated via Expressed Protein Ligation (EPL) to a recombinant polypeptide comprising multiple antigens of Group A Streptococcus [29]. Following subcutaneous administration in mice, the Pam_2_Cys vaccine yielded the highest titre of IgG specific for multiple epitopes, with broader cross-reactivity to a diverse panel of strains than the other micelle formulations tested. Even though our lipid did not harbour all the important features of Pam_2_Cys, it is possible that CD0873-MalLipo generated antibodies with broader epitope recognition than CD0873 that enhanced their binding to CD0873 exposed on the surface of cells of *C. difficile*. Supporting the findings of Kovacs-Simon et al. (2014) [54] we previously showed that CD0873 is indeed abundant on the surface of cells of *C. difficile* 630 [35], the strain used in this study. Pam_2_Cys is not only a potent immunostimulatory adjuvant in parenteral vaccines, it has shown encouraging efficacy in mucosal vaccines, including in an intra-nasal vaccine [28] and in an oral vaccine [55]. In addition to utilising Pam_2_Cys, we plan to conjugate other antigens to our liposomal platform, such as immunological domains of the two toxins of *C. difficile*, and to test this multivalent vaccine for its protective efficacy against *C. difficile* infection in the hamster lethality model.

To summarise, mucosal vaccines negate the need for needles, reducing cost and potentially leading to increased compliance and reduced risk of spread of transmissible diseases, as has been experienced with hepatitis C and HIV following the use of injected vaccines [32]. Furthermore, as strongly reinforced by the COVID-19 pandemic, there is an urgent need for vaccines that can be stored at room temperature and easily distributed globally without the need for refrigeration. Oral vaccines are attracting great interest since they can induce systemic immune responses as well as local and distal mucosal responses, and thus potentially target most body sites. When administered orally in enteric capsules for targeting the small intestine, we show our liposome platform is an effective delivery system for whole-protein antigens, enhancing their immunogenicity. We show that our liposomal formulations can be lyophilised and stored stably at room temperature and reconstituted *in vivo* to induce mucosal and systemic antibody responses without causing any detectable immunopathology. We believe our safe subunit approach could be a preferred oral platform over the current whole cell approach, which remains unsuitable for the elderly or immunocompromised.

## Data Availability

Not applicable.

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
