# Peer review of "Mimicking Native Display of CD0873 on Liposomes Augments Its Potency as an Oral Vaccine against Clostridioides difficile"

_vaccines, 2021, doi:10.3390/vaccines9121453_

Round 1

Reviewer 1 Report

The Authors evaluated the efficacy of a novel oral vaccine against Clostridium difficile to drive intestinal and systemic immune responses. The vaccine that is constituted by the antigen protein CD0873 displayed on liposomes, is administered in enteric capsules for targeting small intestine.

The subject is according to the scope of the Journal. The chosen topic is of scientific interest. The use of English language is appropriate.

I do not have any comment to do and any suggestion for the Authors.

Consequently, I have accepted the manuscript in its form.

Author Response

We thank Reviewer for their endorsement of the manuscript and its acceptance in its current form.

Reviewer 2 Report

Interesting, very methodically advanced and well-written article that I read in one breath. The methodology is meticulously presented, in my opinion it makes it possible to repeat the experiment without major difficulties. The authors' results are promising and worth publishing. A short and to the point discussion fulfills its function with a vengeance. The figures made by the authors deserve praise, they complement the text well. Minor remarks:

  • It seems justified to present a slightly broader characterization of Clostridioides difficile. I was not entirely sure if this is clinical isolate from the patient? Was it obtained from disease stage or from microbiota? Is there more information on the drug resistance of this bacterial strain? Please consider supplementing this information.
  • The good impression of the precisely performed results is spoiled by Fig. 5A. Interpretation is guessing rather than reading the result. Maybe this photo could be corrected before publication?
  • The text should be reviewed editorially. In several places I noticed, for example, the lack of italics next to Latin species names, e.g. line 172: Clostridioides difficile in italics, line 212, 467: E. coli in italics

Author Response

We thank Reviewer  for their thorough review of the manuscript, their endorsement and very helpful suggestions.

  • Point 1 It seems justified to present a slightly broader characterization of Clostridioides difficile. I was not entirely sure if this is clinical isolate from the patient? Was it obtained from disease stage or from microbiota? Is there more information on the drug resistance of this bacterial strain? Please consider supplementing this information.

  • Response 1

In section 2.1, page 5 “Bacterial strains”, the sentence “Clostridioides difficile strain 630 (PCR ribotype 012) was kindly provided by Peter Mullany, UCL”, has been elaborated. This now reads as follows and the relevant citations added to the reference list.

Clostridioides difficile strain 630 was kindly provided by Peter Mullany, UCL. Strain 630 is a virulent, multidrug-resistant strain isolated in 1985 from a hospital patient with severe pseudomembranous colitis which spread to other patients on the same ward in Zurich, Switzerland [47]. This outbreak strain harbours the two toxins, TcdA and TcdB, and belongs to PCR ribotype 012, and is now adopted as the reference strain for laboratory studies [48].

  • Point 2 The good impression of the precisely performed results is spoiled by Fig. 5A. Interpretation is guessing rather than reading the result. Maybe this photo could be corrected before publication?

  • Response 2

The relevant bands on the Western blot image (Fig 5A) have now been quantified for intensity using ImageJ and the data plotted as a histogram, to replace the original image. The following sentence has been added to Methods section 2.14 page 9 accordingly and the relevant citation added to the reference list. Band intensity was quantified using ImageJ calibrated to perform optical density based on a pallet of colours in grayscale [50].

The description of the procedure in Results section 3.6, page 15 has been changed with the following wording deleted,

and replaced with:

An immuno-reactive band of the expected size [51]was observed for all hamsters vaccinated with either CD0873 or CD0873-MalLipo unlike the negative control groups. The intensity of each band was quantified using ImageJ and the results plotted for each group as a histogram. IgA abundance was greatest for the CD0873-MalLipo group with p=0.039 compared to the naïve group.

Relevant changes were made to the figure legend for Figure 5 accordingly.

  • Point 3 The text should be reviewed editorially. In several places I noticed, for example, the lack of italics next to Latin species names, e.g. line 172: Clostridioides difficile in italics, line 212, 467: E. coli in italics

  • Response 3

The text has now been carefully reviewed and changes made to correct typos, ensure italics are used for all names of organisms and enzymes and grammar improved for clarity where needed. The main editorial change for grammar/clarity has been for the first paragraph of the Discussion.

 A revised manuscript with tracked changes has been uploaded.